# A Generalized Threat Model for Visual Sensor Networks

**DOI:** 10.3390/s20133629

**Published:** 2020-06-28

**Authors:** Jennifer Simonjan, Sebastian Taurer, Bernhard Dieber

**Affiliations:** 1Georgia Institute of Technology, Atlanta, GA 30318, USA; jennifer.simonjan@aau.at; 2Joanneum Research, 9020 Klagenfurt, Austria; sebastian.taurer@joanneum.at

**Keywords:** visual sensor networks, security, threat model, distributed systems

## Abstract

Today, visual sensor networks (VSNs) are pervasively used in smart environments such as intelligent homes, industrial automation or surveillance. A major concern in the use of sensor networks in general is their reliability in the presence of security threats and cyberattacks. Compared to traditional networks, sensor networks typically face numerous additional vulnerabilities due to the dynamic and distributed network topology, the resource constrained nodes, the potentially large network scale and the lack of global network knowledge. These vulnerabilities allow attackers to launch more severe and complicated attacks. Since the state-of-the-art is lacking studies on vulnerabilities in VSNs, a thorough investigation of attacks that can be launched against VSNs is required. This paper presents a general threat model for the attack surfaces of visual sensor network applications and their components. The outlined threats are classified by the STRIDE taxonomy and their weaknesses are classified using CWE, a common taxonomy for security weaknesses.

## 1. Introduction

As sensor networks become more and more ubiquitous these days, surrounding us in our everyday environment, the concern for security and privacy increases steadily. The distributed nature of sensor networks and their deployment in remote areas make them vulnerable to numerous security threats that aim at affecting their proper functioning. An important concern is thus to ensure robust and reliable applications even in the presence of attackers. Attackers might want to intrude the network in order to get access to sensitive data or to spread false information through the network. Therefore, they need to be detected as effective as possible on all layers of the sensor network. Defending against attackers and malicious behavior in a potentially large-scale, fully distributed and autonomous network is non-trivial since nodes do not have any previous knowledge about others and solely rely on sensed or received information.

With the growing popularity of IoT applications such as smart homes and industrial systems, the first waves of cyberattacks have been performed. Very prominently, the Mirai botnet [1,2] was used to spread malicious software to millions of IoT devices. In general, Mirai infects IoT devices via remote access, turning them into bots for executing Distributed Denial of Service (DDoS) attacks. This incident was an eye opener, showing how big the IoT insecurity problem is. In 2017, Bricker bot attacked around 10 million home routers and IP cameras before it was discovered by Radware [3]. This malware targets IoT devices to perform Permanent Denial of Service (PDoS) attacks attempting to permanently destroy insecure devices by degenerating their storage. Very drastic privacy invasions on the vulnerabilities of baby monitors have also been reported lately [4].

Many more reports have been made public throughout recent years about the vulnerabilities of dedicated IoT devices. Summarized, there are many different application domains which can be affected through IoT security breaches in order to attack the privacy of humans or the proper functioning of systems. These applications include, e.g., nuclear facilities, energy grids and water supplies, hospitals and medical systems, building infrastructure, airplanes and autonomous cars and private homes. As sensor nodes become smaller nowadays, eventually reaching sizes in the scale of nanometers [5], they will be invisibly embedded in our everyday life environment which poses new challenges on privacy and security requirements, motivating the need for a clear understanding of all potential security threats.

This paper presents a general threat model for the attack surfaces of distributed visual sensor network (VSN) applications and their components. A threat model for a specific system describes the potential ways an attacker can compromise this system. Based on that, a threat analysis can be used to quantify the risk of a certain attack vector and mitigation can be prioritized. Typically, threat models are specific to one system. However, a general threat model like presented in this work may be used to guide the threat modeling activities for a specific system based on the class of application. This threat model extends previous works in the fields of Internet of Things (IoT) and wireless sensor networks (WSNs). Furthermore, common mitigation strategies for the identified attack vectors are presented.

Motivation and contribution: While various threat models for WSNs and IoT can be found in literature, there is no comprehensive and strategical threat model for understanding the holistic nature of VSN security for practical deployments. Due to the increased use of camera networks in our everyday life (i.e., public video surveillance), raising the awareness of security flaws in VSNs is a first important step towards future, privacy preserving camera networks. The main contribution of this paper is a generalized threat model for distributed VSNs that collects common threats in such networks. The model is build according to STRIDE and is based on five different layers, considering potential attacks in the intra-camera as well as the inter-camera domain. It is intended as a guide for engineers in threat modeling their specific VSN.

Outline: The remainder of this paper is organized as follows. Section 2 discusses the related work in security threats and solutions for IoT, SNs and VSNs as special case of SNs. The building blocks used to categorize all threats of the proposed threat model are presented in Section 3. Section 4 presents the general threat model for VSNs and Section 5 briefly discusses mitigation and scoring techniques used to assess the risk of a specific threat. Finally, we conclude our work in Section 6.

## 2. Related Work

The rapid growth of sensor networks (SNs) and the Internet of Things (IoT) paradigm have induced a huge attention from the research community. Sensor and IoT networks have several security vulnerabilities which can be exploited by attackers to launch various types of attacks. Therefore, much research has been focused on identifying potential security threats as well as solutions in these networks. The following subsections discuss related work in security threats and solutions for IoT, SNs and VSNs as special case of SNs.

Internet of Things: The IoT is rapidly growing due to the proliferation of communication technology and the availability of the devices and computational systems. Application examples of existing IoT systems include self-driving vehicles for automated vehicular systems, microgrids for distributed energy resource systems and Smart City Drones for surveillance systems [6]. Various surveys have been presented throughout the last decade addressing IoT-centric topics such as intrusion detection systems, threat modeling and emerging technologies [6,7,8,9,10,11,12,13,14]. The two most recent surveys were presented in 2019 by Hassan et al. [6] and Neshenko et al. [10]. Both works present an in-depth overview of security flaws in IoT. Neshenko et al. [10] additionally includes an extensive discussion of the state-of-the-art in IoT security countermeasures. Another survey on threats in the IoT was presented by Alaba et al. [11]. The paper discusses the state-of-the-art IoT security threats and vulnerabilities and presents a threat taxonomy categorized by application, architecture and communication. Angrishi et al. [12] and Hodo et al. [13] have also presented studies on IoT vulnerabilities. The latter additionally proposes a solution which comprises an Artificial Neural Network (ANN) to combat the threats. Sfar et al. [9] have presented a roadmap for security challenges in the IoT, in which they survey security related interactions and solutions based on privacy, trust, identification and access control. Another comprehensive overview of the IoT paradigm with respect to system architecture, enabling technologies and security and privacy issues has been presented by Lin et al. [8]. The authors further present the integration of fog and edge computing with IoT. Yang et al. [7] have presented a survey categorized by four segments: (1) The most relevant limitations of IoT devices and their solutions, (2) the classification of IoT attacks, (3) the mechanisms and architectures for authentication and (4) the access control and security issues in different layers. Regarding security solutions, Kouicem et al. [15] have suggested traditional cryptography methods and new technologies such as Software Defined Network (SDN) and Blockchain [16] to be implemented to solve current IoT security issues.

Sensor Networks: The fact that autonomous networks like sensor networks have a high exposure in terms of security risks has been discussed in related work for several years [17] with the earliest works dedicated to sensor network security appearing around 2002 [18,19]. Similar to IoT, there is also a large variety of surveys and studies on security threats and solutions in SNs [20,21,22,23,24,25]. A comprehensive classification of security attacks in wireless SNs has been introduced by Padmavathi et al. [20]. After categorizing and discussing potential attacks, the authors introduce security mechanism to detect, prevent and recover from these attacks. They further summarize the main sensor network security challenges to the following: Wireless medium, ad-hoc deployment, hostile environment, resource scarcity, immense scale, unreliable communication and unattended operation. A strategical threat model which categorizes threats into three main classes, namely outsider attacks, insider attacks and key-compromise attacks, was introduced by Cardenas et al. [21]. The authors discuss all kinds of attacks considered in their threat model as well as the security design space for the use case of Supervisory Control and Data Acquisition (SCADA) systems. A dedicated threat model for ad hoc sensor networks has been presented by Clark et al. [22]. Their model also indicates components that can be used to form an adversary model. The authors identify threat categories, modes of use and a variety of threats to the system assets including threats to communications, infrastructure services, individual nodes and threats concerning the human element. Another threat model has been introduced by Westhoff et al. [25]. The authors describe a dedicated threat model as well as security solutions for collecting and processing data in wireless SNs. The paper includes an overview on security and reliability challenges for wireless SNs and introduces a toolbox concept to support such a framework. Futher works dealing with security issues in SNs have been presented in [24,26,27]. Grover and Sharma [26] study the various security issues and threats in wireless SNs based on the different layers and summarize some well-known protocols used to achieve security in the network. Sinha et al. [27] also study security vulnerabilities, followed by a classification of various attacks according to different OSI protocol layers. In [24], the authors focus on discussing physical attacks in wireless SNs and on identifying the purpose and capabilities of attackers. Furthermore, they discuss well-known detection approaches for physical attacks.

Visual sensor networks: VSNs are a special case of SNs as they pose additional security vulnerabilities due to the multimedia contents which are processed and transmitted through the network. Camera networks are used in many contexts nowadays and thus their application domains are continuously growing. Example applications include indoor/outdoor surveillance, traffic monitoring, assisted living for disabled or elderly people, localization and recognition of services and users, environment monitoring and industrial monitoring [28]. Having access to sensitive and potentially private image data results in an even higher need for security and privacy solutions. However, compared to traditional SNs and IoT, there is only little research regarding strategical threat models for VSNs. One security survey has been presented in 2014 by Winkler et al. [29]. The authors discuss the major security threats and attack scenarios of VSNs classifying the security aspects into data-centric, node-centric, network-centric and user-centric security. Furthermore, privacy protection techniques for smart cameras are discussed. Another study was introduced by Kundur et al. [30], who present issues in the design of security and privacy in distributed multimedia sensor networks. The authors introduce the HoLiSTiC paradigm, which stands for Heterogeneous Lightweight Sensornet for Trusted visual Computing and encompasses the salient features of many proposed multimedia networks in the research literature. The paradigm classifies nodes into base stations, transport nodes and camera nodes and the authors discuss the security challenges and requirements for each type. Grgic et al. [31] analyze security requirements of visual sensor networks and describe possible security threats and attacks, while also suggesting possible countermeasures. The authors additionally propose a security framework for potential VSN applications. An overview of secure VSNs was presented by Grieco et al. [28] in 2009. The authors have also forecasted future perspectives of secure VSNs, outlining management of QoS and QoE, privacy, trust management, integrated platforms and nanotechnology. In 2015, a survey of the security of image-based wireless sensor networks has been presented [32]. The paper focuses on security measures that can be applied in image processing despite the limited resources available at the sensor nodes. Furthermore, there are several works which present solutions to combat specific security issues in VSNs [33,34,35]. These include, i.e., watermarking of images, aggregation and comparison of multimedia data from different nodes and cryptography to secure the transmission of images.

A special network type are heterogeneous VSNs exploiting not only cameras but different kinds of sensors. The literature research shows, that similar to homogeneous VSNs, there is just little work in security for heterogeneous VSNs. One approach, proposed by Kunst et al. [36], deals with the secure transmission of surveillance videos in smart city applications by implementing QoS-aware resource sharing in heterogeneous network scenarios. Another approach was introduced by Zhou et al. [37]. The work includes a security-critical multimedia service architecture, which jointly considers traffic analysis, security requirements, and traffic scheduling for IoT multimedia applications.

As already mentioned in the introduction, the literature research shows that there is a lot more work regarding security threat analysis in wireless SNs and IoT than in VSNs. Specifically, the state-of-the-art does not include any comprehensive threat model for VSN security yet. Thus, this paper establishes a comprehensive threat model for distributed VSNs, considering all layers in the intra- and inter-camera domain. The threat model can only be partly applied to heterogeneous VSNs, since it does not consider any other sensor types than cameras. In the case of a heterogeneous network, additional layers of attack for the different physical platforms and their specific software are required.

## 3. Threat Model Building Blocks

This section describes the building blocks used to categorize all threats in the proposed threat model (which is presented in Section 4). First, all layers of a typical distributed VSN application are explained in order to be able to establish the threat model in a layer-wise fashion. Section 3.2 introduces the STRIDE paradigm, which is used to categorize threats into specific groups. Section 3.3 discusses the common weakness enumeration (CWE), which is a scoring system for security tools. Last, a simplified attacker model, which enables to classify attacks by the level of knowledge required by an attacker to perform it, is presented.

### 3.1. Visual Sensor Network Model and Threat Model Layers

The proposed threat model is based on a generalized model of a visual sensor network. Cameras are assumed to be enclosed, standalone devices connected to visual sensors. On the computing device, an operating system like Linux is used to abstract the underlying hardware. The visual sensor is accessed via an operating system driver. The VSN business application is running as one or multiple processes on top of the operating system, accessing the visual sensor via a driver and potentially using other system parts like embedded GPUs or IO. To realize a distributed application, a middleware like MQTT, ROS or similar is used.

To explore all kinds of attacks on a visual sensor network, the different layers of a sensor node that could be attacked are explored first. Figure 1 depicts an overview of the attack layers of networked visual sensor nodes. The outermost layer is the network, which enables the nodes to communicate with each other (i.e., through a middleware system) or to connect to the internet. Intruders might want to attack the network layer of a VSN by a variety of attacks such as jamming, eavesdropping, traffic analysis or manipulation of frames. The physical node itself, comprising all physical components (i.e., image sensor) and connections (i.e., USB), forms another vector of attack. Physical node attacks include, i.e., node capture, destruction or displacement and (dis-)connection of components. A further layer of attack is the operating system, which bridges the hardware and the software of the sensor node. Thus all kinds of drivers and services can be attacked via the operating system. The last potential layer of attack is the distributed application running on top of the operating system and performing the actual task of the VSN. Attacks on the application typically aim at manipulating the operation of distributed algorithms or at tampering data flows in order to falsify, i.e., localization, synchronization or configuration data.

### 3.2. The Stride Threat Modeling Technique

STRIDE stands for Spoofing, Tampering, Repudiation, Information Disclosure, Denial of Service and Elevation of Privilege and represents an approach to threat modeling, which was invented by Kohnfelder and Garg [38]. This framework was designed to support developers in identifying attack types that their software could experience [39] and is a core part of the Microsoft Security Development Lifecycle [40]. The STRIDE threats represent the opposite of the standard security properties that software systems are desired to have, namely authenticity, integrity, non-repudiation, confidentiality, availability and authorization.

Table 1 shows the STRIDE threats along with their definition and the violated security property. The first threat shown in the table is spoofing, which occurs when an attacker successfully impersonates or masquerades a legitimate user, process or system element. Authentication aims at avoiding spoofing attacks by proving an entity’s identity. The second threat is tampering, which refers to the modification of legitimate data or code. Tampering is tackled by integrity, which involves maintaining the consistency, accuracy and trustworthiness of data over its entire life cycle. The next STRIDE threat is repudiation, which occurs when an entity denies an action and it cannot be proven that the action was performed. In order to cope with repudiation attacks, assurance that entities cannot deny their actions is required. This property is referred to as non-repudiation. The fourth STRIDE threat is information disclosure, which occurs when information is exposed to an unauthorized entity. The security property tackling information disclosure is confidentiality, which ensures that information is always kept private. The next threat comprises Denial of Service attacks, which aim at denying, degrading or interrupting any service to valid users. Availability is required in order to ensure that authorized parties are able to access the information when needed. The last STRIDE threat is elevation of privilege, which occurs when an entity gains higher privileges in the system without being authorized to do so.

Summarized, STRIDE is a method for structured analysis of a system or component that aims at identifying vulnerabilities which could be exploited by an attacker to compromise the whole system [41]. It provides a structured way of categorizing threats in order to determine strategies to mitigate them and is thus typically used in the design of secure software systems.

### 3.3. Common Weakness Enumeration

The Common Weakness Enumeration (CWE) (https://cwe.mitre.org) is a community-developed list for software and hardware flaws and vulnerabilities. As such, it serves as a common language, a scoring system for security tools and as a baseline for weakness identification, mitigation, and prevention efforts. CWE categorizes common weaknesses that can occur in a system and relates the weaknesses to hierarchies. As such, it does not name specific vulnerabilities in specific systems (such as the related Common Vulnerability and Exposures (CVE) project (https://cve.mitre.org) does) but assigns IDs to classes of vulnerabilities. Using these IDs, weaknesses found in specific systems can be related, compared and classified. Based on that, more information on impacts and mitigations can be obtained. Furthermore, CWE entries are often linked to specific known vulnerabilities in the CVE list.

The CWE suggests a certain hierarchy, which is composed of classes, bases and variants. Classes are described in a very abstract fashion, typically independent of any specific language or technology. Bases are also described in an abstract fashion, but with sufficient details to infer specific mitigation strategies. Variants are linked to a certain type of product, typically involving a specific language or technology. Moreover, CWE can be presented from different views, namely (i) by a research concept (which is abstract and presents a complete CWE list), (ii) by software development or (iii) by hardware design. The software and hardware concepts are both subsets of the complete CWE list. The research concept includes one more hierarchy layer, namely pillars. Pillars are the highest-level weakness and top-level entries in the research concepts representing an abstract theme for all class/base/variant weaknesses related to it.

For better understanding, consider the following example derived according to the bottom-up research concept: CWE-520 defines the variant. “NET Misconfiguration: Use of Impersonation” which is a child of base CWE-266 “Incorrect Privilege Assignment” which in turn is a child of class CWE-269 “Improper Privilege Management”. The top-level entry, and thus the pillar, for this example is CWE-284 “Improper Access Control”.

This work provides the appropriate CWE IDs for threats of the proposed threat model. The ID can then be used to gain more insight into known instances of the weakness, potential impacts and mitigation techniques. It is important to notice, that not all threats of the proposed threat model can be mapped to a CWE ID. This is due to the fact, that VSNs are cyber–physical systems, which are also exposed to non-digital attacks in the physical world.

### 3.4. Attacker Model

Since the presented threat model for VSNs is generalized, the assumed attacker model is a simplified one. Typically, one would consider insider vs. outsider and different organizational levels of attackers [42]. It is strongly recommended to do this in an application-specific threat model where the accompanying and organizational circumstances are known.

All security attacks are classified by the level of knowledge required by an attacker to perform it. Thus, we take a look on how skilled an attacker has to be in order to perform a certain kind of attack. The three levels of knowledge considered for the classification are the following:Amateur: No specific knowledge or equipment is required to perform the attack. The attacker may be an ordinary user who “stumbles” across a vulnerability. Examples for amateur attacks are node capture or destruction, since the attacker does neither require specific technical knowledge nor dedicated equipment to steal or physically destroy a node. Physical Denial of Service attacks, such as blocking a camera’s view or disconnecting components also belong to this category, since they can be easily performed.Skilled: Technical skills and the ability to understand and manipulate the working environment is required to perform the attack. In this level, we assume the attacker to be able to acquire enough domain knowledge for more specific attacks. Examples for attacks which can be performed by skilled attackers include stealing keys, modifying software or eavesdropping the communication. To be able to perform these attacks, the attacker definitely requires technical domain knowledge.Expert: Very specialized knowledge and dedicated equipment is required to perform the attack. The attacker is assumed to have domain expertise, i.e., is proficient in Visual Sensor Networks and methods like computer vision, middlewares systems or artificial intelligence methods. Such attacks may even require the attacker to be an interdisciplinary team. Example attacks include the manipulation of the network calibration or synchronization, AI bias attacks in which attackers aim at influencing AI algorithms to falsify their results, and the manipulation of the sensor node clock or CPU performance.

## 4. A Generalized Threat Model for Visual Sensor Networks

This section presents a general threat model for visual sensor networks. The model aims at identifying the threats that will be present in almost all applications of this technology. It is important to notice, that threat modeling should always be done specific to an application and its configuration. Depending on the application scenario, the hardware, operating system and software of a specific visual sensor network, new threats may occur (e.g., attacks specific to the version of an operating system) and others may not be present (e.g., operating system attacks in a true embedded system). The proposed threat model should be used as a guide for the specific threat modeling process. Developers are strongly encouraged to include threat modeling in their development process and to mitigate all the identified threats as early as possible.

### 4.1. Threat Model and Attack Vectors

The proposed threat model aims at covering visual sensor networks in the best sense of distributed systems. Thus, it fundamentally distinguishes between intra- and inter-camera attack vectors. Figure 2 shows the classification of security attacks in visual sensor networks. First, it is assessed if a certain attack vector is aimed at a single camera or at the whole networked application. The intra-node attack vectors distinguish between physical attacks on the node and attacks on its software including the operating system as well as the business software. The inter-camera domain further divides attack vectors into attacks on the network, on the middleware and on the distributed algorithm (the application code) itself. Every attack is additionally classified by its STRIDE threats (depicted by the red uppercase letters in the figure), the CWE ID and the level of knowledge required to perform the attack.

### 4.2. Intra-Camera Threats

This section discusses intra-camera threats attacking the sensor nodes physically and the software (operating system and business software) running on the nodes.

#### 4.2.1. Physical Attacks

This subsection describes attack vectors that emit a physical influence on the host. For all cases, the attacker needs to be in physical proximity in order to carry out the attacks. Gathering physical access to sensor nodes in uncontrolled or unobserved environments is typically easy for attackers. Figure 3 depicts an overview of the physical layer weaknesses, which are mapped to their respective impact. Since the OSI model only characterizes the communication functions of computer systems, it was extended by two additional layers, namely the node software layer and the physical node layer. The layer affected by physical attacks is the physical node layer which is depicted in green in Figure 3. Note that physical attacks which are non-digital can not be mapped to a CWE ID but only approximated with CWE-693 “Protection Mechanism Failure”, which by itself was originally not meant to describe missing physical protection.

Node capture: Nodes in VSN applications are typically easily accessible due to unattended physical locations. An attacker could thus capture one or more sensor nodes to, e.g., achieve a Denial of Service, extract cryptographic keys, modify software or replace the node by a malicious one which is under control of the attacker [20]. Furthermore, attackers could perform side channel attacks which exploit characteristics of the circuit such as timing and power consumption [29]. Node capture attacks are thus either tampering or Denial of Service threats when classified by STRIDE. The level of knowledge required to perform a node capture depends on the goal of the attacker. In the simplest case of just stealing the node, an attacker does not need any technical knowledge or dedicated equipment and can thus be an amateur. If the attacker wants to extract keys or modify the software, skilled knowledge is at least required.

Destruction: Attackers might physically destroy the sensor, processor or components that they depend on thus rendering the node inoperable [43]. This kind of attack is typically easy to detect since the node simply stops working. According to the STRIDE model, it is a Denial of Service attack. Furthermore, destruction is also easy to launch and can be achieved by amateurs.

Displacement: An attacker might re-locate or rotate the sensor thus invalidating the sensor measurements or the network topology and/or coverage [21]. This might result in improper functioning of the application and is thus an effective attack against service integrity. Displacement might either degrade services required for a proper functioning of the application or might aim at (non-)coverage of specific areas. Thus, a displacement of a visual sensor node can be categorized as Denial of Service or tampering attack. Furthermore, displacement of nodes does not require an attacker to have technical knowledge or equipment.

Visual disturbance: An attacker may interfere with the sensor’s perception itself by blocking its view partly or in total. This leads to false detections or no coverage in certain areas. Manipulation of the field of view (FOV) of a camera node can be easily achieved by, i.e., covering the sensor or placing large objects in front of the node. The attack might aim at denying or degrading services which could lead to improper functioning of the application. However, it might also aim at manipulating the view of the visual sensor node by covering just a certain part of the FOV. Visual disturbance is classified as Denial of Service or tampering attack. It does not require specific knowledge or equipment and can thus be achieved by amateurs.

Disconnect components: By disconnecting the visual sensor, the power supply or any other vital component, an attacker can de-facto disable the whole node. Similarly, disconnection of power supply or wired network connections can disrupt the node’s operation. Through the disconnection of different components an attacker can deny any kind of service resulting in a Denial of Service attack. This attack can easily be performed by amateurs.

Connect components (CWE-1203): The attacker may connect additional components like a new visual sensor, external memory or input devices. This would allow the attacker to inject false data if the application software is not protected against that (CWE-346) or gain control over the node itself. Exposed interfaces like UART or JTAG (CWE-1191) can easily become gateways for attackers [4]. Due to their intended use as debugging interfaces, they do typically not include any security measures like authorization. Thus, if they are physically accessible to an attacker, this can grant the attacker full control over the system. Furthermore, attackers might want to perform hardware person-in-the-middle attacks by plugging a device in between two communication parties in order to intercept the communication [44]. Therefore, connecting components classifies either as spoofing or as tampering attack, depending on if the attacker aims at impersonating the node or at modifying data or code. The level of knowledge required by an attacker to perform these kinds of attacks reaches from amateur to skilled, depending on the specific goal.

Manipulate power supply (CWE-1206): Besides disconnecting the power supply of the node, the attacker may manipulate the voltage (CWE-1247) to make the system unstable or provoke a higher error rate in computation (assuming improper voltage regulation on an embedded device) [45]. Furthermore, an attacker might want to perform side channel attacks via power-analysis in order to get access to system-relevant data such as private keys. Thus, manipulation of the power supply is classified as Denial of Service or tampering attack. An attacker is required to have at least skilled knowledge in order to perform such an attack.

#### 4.2.2. Software Attacks

Besides physical attack vectors, software-driven attacks on a single node are also foreseen. These include attacks at the operating system level as well as manipulations of the business software running on it. Figure 4 gives an overview of the software layer weaknesses, including both the operating system and the business software attacks. These attacks are mapped to the affected extended-OSI layer, namely the node software, and to their respective impact.

##### Attacks on the Operating System

Gaining root (CWE-269): The premier goal of an attacker pursuing a software attack is to receive root or administrator rights on a host. This is typically done via privilege escalation [46] and is thus classified as elevation of privilege attack. To be able to perform this attack an attacker needs at least skilled knowledge.

Manipulate drivers (CWE-782): If an attacker manages to manipulate the sensor drivers of a VSN node, this may become a channel to inject forged image data without being recognized by the application. Analogue, this may be used to eavesdrop data. Similar intents could be achieved with drivers for other devices. A fake driver module may therefore be used to gain unauthorized access to data, to modify data, to spoof connected devices or deny their services. A manipulation of the drivers requires attackers to be skilled.

Manipulate clock (CWE-1247): An attacker might want to manipulate the clock of the sensor nodes in order to alter the timestamp of transmitted packets [47]. A manipulated clock leads the sensors to capture their readings at wrong times. Furthermore, it destroys the time synchronization of the whole network. The attack aims at manipulating data and is thus classified as tampering attack. An attacker needs to have expert knowledge in order to be able to manipulate the clock of a sensor node.

Manipulate network connection (CWE-285, CWE-668, CWE-732): An attacker may manipulate the operating system configuration of the networking stack. This includes the manipulation of routes and nameservice entries, general disconnection or the connection of a wireless interface to a false network. The attack is therefore classified as tampering or Denial of Service. Once appropriate privileges are gained, this attack can be performed by a skilled attacker.

Manipulate hardware performance (CWE-1201): An attacker could manipulate the hardware performance by, e.g., lowering the CPU performance in order to slow down the system (e.g., reducing the image framerate) or to completely stop algorithms from working. The attack might either aim at manipulating data or at denying services in order to disturb the functionality of the application. It is thus categorized as tampering or Denial of Service attack. To be able to manipulate the hardware performance, an attacker needs to have expert knowledge.

Attack memory (CWE-284, CWE-922): An attacker might want to attack the memory in order to steal relevant information or make the memory unavailable for the node. Specifically, an attacker could limit the available memory by occupying it with random data or eavesdrop the memory (CWE-316) to get access to secret or private data such as keys (e.g., copying the memory content). Similarly, false data could be injected to the memory to manipulate the application accessing it. Attacking the memory can thus be classified as information disclosure, Denial of Service or tampering. To be able to launch memory attacks, an attacker needs at least to be skilled.

##### Attacks on the Business Software

Modify business software (CWE-840): An attacker might want to modify the software running on a node by, e.g., exchanging it or parts of it. An easy entry point for attackers are buffer overflows (CWE-119), which are typically exploited to subvert the function of a privileged program, to run arbitrary code or trigger a response that damages files, changes data or unveils private information [48] (CWE-200, CWE-359). Malicious software injected into the network could spread to all nodes, potentially destroying the whole network or taking over the network on behalf of an adversary [20,49]. Modifying the business software can thus result in modified data or code or in degradation of services and is thus categorized as tampering or Denial of Service attack. The modification of business software requires an attacker to be skilled.

Manipulate configuration (CWE-16): An attacker may manipulate the local configuration of the VSN application in order to disturb its operation. This can comprise the tampering with configuration files, startup scripts as well as local configuration stores like embedded databases. Once access is gained, this attack can be executed by a skilled attacker.

AI bias (CWE-840, CWE-20): An attacker with full access to the software layer might take actions to bias artificial intelligence algorithms running on the node. By feeding forged data into online learning algorithms or classifiers, the underlying model of the algorithm can be manipulated [50] (CWE-1039). The potential effects on vision recognition algorithms has been discussed in [51]. One famous example of AI manipulation is the Tay chatbot by Microsoft [52]. A further attack vector in case of modular AI pipelines [53] is exchanging individual modules with forged ones. Such attacks aim either at manipulating data or algorithms, or on degrading services for the business software. Thus, they are classified as tampering or Denial of Service attacks. An attacker needs to have skilled knowledge in order to launch AI bias attacks.

### 4.3. Inter-Camera

This section discusses inter-camera threats attacking the communication in the network and the distributed application of the VSN.

#### 4.3.1. Networking

This subsection introduces attacks that target the network connecting the VSN nodes. Figure 5 gives an overview of the network layer weaknesses, which are mapped to the affected OSI layer (green colored boxes) and their respective impact.

Interference attacks (CWE-400, CWE-406): Attackers might want to disrupt the communication in the network by jamming it. To do so, attackers would broadcast a high-energy signal which disrupts all other network communication [54,55]. If the transmission is powerful enough, the entire system’s communication could be jammed. Another possibility to disrupt communication is to violate the medium access control (MAC) protocol which regulates the transmit permissions of the nodes [20]. Jamming the communication and violating the MAC protocol are Denial of Service attacks, since they keep nodes from communicating with each other or from accessing the channel. An attacker would at least need skilled knowledge to perform interference attacks.

Routing attacks: Routing is a crucial service for enabling communication in sensor networks, however, it typically suffers from various security vulnerabilities [20]. The simplest routing attacks involve the injection of forged routing information (CWE-290) into the network to perform, e.g., spoofing, replaying or alternation of routes [56]. Furthermore, attackers might want to attract all the traffic to eventually drop it (e.g., black hole attack), to create fake identities in the network to outvote benign nodes (e.g., Sybil attack) or to drop packets selectively to deteriorate the network efficiency (e.g., selective forwarding attack) [30,57,58]. The various routing attacks fall into the categories of spoofing, tampering or Denial of Service attacks. Depending on the specific routing attack, an attacker requires either skilled or expert knowledge to launch it.

Inject frames (CWE-924): An attacker might want to cause unreliable transfers, packet collisions or latency on purpose in order to harm the proper functioning of the VSN. For example, achieving a time synchronization within the network is barely possible in the presence of significant latency [59]. Furthermore, attackers could alter the content of a message thus destroying its integrity. By injecting manipulated frames into the network, an attacker might either aim at impersonating a node or at spreading false information. The attack can thus be classified as spoofing or tampering attack. To launch such an attack, the attacker requires at least skilled knowledge.

Manipulate key management (CWE-320, CWE-322): The main goal of key management is to establish a key exchange between sensor nodes safely and reliably [32]. An attacker might want to eavesdrop, steal or manipulate keys (or key exchange) in order to get access to the information transmitted within the network. The attack is thus classified as information disclosure or tampering. An attacker needs to have at least skilled knowledge in order to launch such an attack.

Eavesdropping (CWE-300, CWE-319): Eavesdroppers overhear the communication in the network in order to gather confidential data or learn about secret keys and/or identities [60] (CWE-523). Besides passive overhearing, attackers might even perform active eavesdropping by sending queries to sensors or aggregation points. Eavesdropping is an information disclosure attack which requires an attacker to have skilled knowledge.

Denial of Service (DoS) attacks: DoS attacks can in general be launched on different layers to target the availability of services provided by the network or the nodes. At the networking layer, DoS attacks include packet collision, exhaustion due to repeated transmission, misdirection, black holes, unfairness in using the wireless channel, malicious flooding (CWE-406) and de-synchronization [21,61]. To launch DoS attacks on the networking layer, an attacker needs to be skilled.

Person-in-the-middle attacks (CWE-300): In a person-in-the-middle attack, benign nodes communicate via an attacker without being aware of it. The attacker has typically complete control of the communication link which enables to deny services and to delay, delete, modify or re-order the traffic in order to trick the receiver into unauthorized operations such as false identification or authentication [62,63]. Person-in-the-middle attacks affect the communication between two parties by modifying data and are thus classified as tampering attacks. In order to launch a person-in-the-middle attack, the attacker needs to have at least skilled knowledge.

Replay attacks (CWE-294): A malicious node performing a replay attack could store information without authorization to later retransmit it in order to fool the receiver. For example, an attacker might retransmit a network-login message of a benign node at a later point in time. Even if the message is encrypted and the attacker is not aware of the keys or passwords, the retransmission of a valid login message might enable the attacker to gain access to the network [63]. Additionally, an attacker could forge the source of the replayed message. Since replay attacks aim at impersonating other network entities, they are categorized as spoofing attacks. An attacker requires at least skilled knowledge in order to successfully launch replay attacks.

Traffic analysis (CWE-497): An attacker might want to monitor the network traffic in order to learn about computing parameters which can affect the network. Packet traffic in a sensor network typically shows certain patterns which allow attackers to analyze network parameters such as the location of the basestation or aggregation point [64]. Traffic analysis aims at accessing information in an unauthorized way and is thus classified as information disclosure attack. Monitoring and analyzing network traffic require an attacker to have skilled knowledge.

#### 4.3.2. Middleware

This subsection describes attack vectors on the underlying middleware used in a VSN application. Such attacks may exploit weaknesses of middleware paradigms or implementations and may not be considered by an application developer. Figure 6 depicts an overview of the middleware layer weaknesses, which are mapped to the affected OSI layers, namely session and presentation layer, and their respective impacts.

Manipulate communication graph (CWE-923): An attacker can modify the communication flow of the middleware by connecting previously unconnected nodes, isolating nodes or introducing new (malicious) nodes. As an example, attacks like those have been presented for the publish/subscribe middleware ROS (Robot Operating System) [65,66]. Manipulating the communication is a tampering attack, which can in the case of node isolation or disconnection also result in a Denial of Service. Attackers need skilled knowledge in order to manipulate the communication graph on the middleware layer successfully.

Inject data (CWE-940): An attacker could inject forged data into an application by publishing data for arbitrary topics without prior authorization [67]. Injecting manipulated commands or sensor readings could lead an application to fail or even to perform in a harmful way. Within a VSN application, manipulated data may include forged images, false processing results, fake object detections and other kinds of falsified data. Injecting forged published data is classified as spoofing or tampering attack. To harm the application significantly, an attacker needs to be aware of the used middleware system, the publish/subscribe topics and the purpose of the application and requires thus skilled knowledge.

Manipulate configuration (CWE-16): Many middleware systems take care of managing the parameters of the distributed application, i.e., through a parameter server. One example for such a server is the ROS parameter server (http://wiki.ros.org/Parameter%20Server), which is a shared, multi-variate dictionary that is accessible via network APIs. Nodes use this server to store and retrieve parameters (typically configuration parameters) at runtime. In such a setting, an attacker would have to attack the middleware layer in order to manipulate the configuration. Modifying parameters is considered as tampering attack and would require an attacker to be skilled.

Manipulate key management (CWE-320): The middleware system might be capable of cryptographic methods in order to store or transmit data in a secure way. If so, an attacker would need to steal or manipulate the keys on the middleware layer rather than on the networking layer. The manipulation of keys is considered as tampering attack and requires an attacker to be skilled.

Eavesdropping (CWE-300, CWE-941): With unauthorized access to the communication flow of an application, an attacker can eavesdrop data from the communication within the VSN application. This can simply be achieved by subscribing to all topics of interest such that all published data is received. This attack is considered as information disclosure, since information is exposed to an entity which is not authorized to see it. For that purpose, attackers need to be aware of the middleware system and the publish/subscribe topics and need thus to have skilled knowledge.

DoS attacks (CWE-406): Publishing a large amount of data can be used as method to take out parts on an application by overloading the hosts’ processing capacities. Further, an attacker could compromise subscription topics to induce periodic authentication challenges or cause isolation via indirect communications through specific topics [68]. Doing so, an attacker is able to deny services essential for a proper functioning of the application. Depending on the middleware system, an attacker needs to be skilled or an expert in order to successfully launch DoS attacks.

Person-in-the-middle attacks (CWE-300): An attacker cannot only inject false data but can also prevent the subscriber from receiving the real data (without noticing it). This also enables person-in-the-middle attacks since a malicious node may act as a subscriber to a publisher and as a publisher to a subscriber and transparently record and manipulate the data flow between those two nodes [65]. Manipulating the communication between two parties is considered as tampering attack and would require an attacker to be skilled or expert, depending on the used middleware system.

Replay attacks (CWE-294): Similar to the networking layer, a replay attack on the middleware layer also exploits retransmission of recorded messages. However, replayed messages are sent in the name (in terms of middleware-ID) of the original, legitimate sender but with the address of the attacker since the source of the message cannot be forged on the middleware layer. The attack is considered as spoofing attack, since it aims at impersonating another node. In order to perform a successful replay attack on the middleware layer, an attacker needs to have skilled knowledge.

Fake/impersonate nodes (CWE-290, CWE-923, CWE-266, CWE-287): An attacker might create fake nodes or impersonate benign nodes through publishing forged data. By publishing data with various node identifiers, the attacker could even pretend to be multiple nodes with, e.g., different locations in order to fool the network. This attack aims at impersonating and faking nodes and is thus considered as spoofing attack. Depending on the middleware system, an attacker needs to be skilled or expert to be able to launch the attack.

#### 4.3.3. Distributed Application

This section discusses attacks on the application, which is typically running in a distributed fashion in modern VSNs. On this layer, an attacker would want to manipulate the operation of distributed algorithms or tamper application data flows. Figure 7 depicts an overview of the distributed application layer weaknesses, which are mapped to the affected OSI layer, namely the application layer, and their respective impacts.

Inject data (CWE-940): Attackers could falsify data on purpose in order to manipulate the proper functioning of the application. Such an attack may be launched by subverted nodes (i.e., formally legitimate participants of the sensor network that have been taken over by an attacker), which already completed the authentication process successfully [69]. Furthermore, attackers might want to inject forged images into the network to stop the computer vision algorithms from working properly. Injecting manipulated data or subverting a node can be considered as tampering or spoofing attack. In order to launch such an attack, the attacker requires expert knowledge.

Manipulate configuration (CWE-16): If the distributed application uses a custom mechanism of managing application-wide configuration parameters (i.e., they are not handled using a middleware-provided mechanism), a parameter manipulation attack similar to that on the middleware layer can be performed on the level of a distributed application. To successfully perform this tampering attack, the attacker needs to have skilled knowledge.

Manipulate intrinsic/extrinsic calibration (CWE-693, CWE-345)): Many VSN applications require an extrinsic and/or intrinsic calibration of the cameras. In the case of extrinsic calibration or localization, attackers may falsify data on purpose in order to manipulate the results. Specifically, they might pretend to be positioned somewhere else to make others believe that a certain sensitive area is covered, while it is not in reality [69]. In the case of intrinsic calibration, an attacker might aim at falsifying the cameras’ intrinsic parameters (i.e., focal length, optical center) in order to stop the application from producing meaningful results. To perform either of these attacks, an attacker needs to manipulate data and thus the attacks are considered as tampering. Furthermore, an attacker needs to be an expert in order to successfully manipulate the calibration of the network.

Manipulate time synchronization (CWE-361): By manipulating the time synchronization of a network, an attacker can prevent it from effectively executing applications which involve collaborative in-network processing (e.g., collective sensing or object tracking). The attacker could even disable the communication in a network by disrupting fundamental services such as the TDMA-based channel-sharing scheme [70]. As in the case of calibration, the manipulation of time synchronization algorithms is also considered as tampering attack requiring attackers to have expert knowledge.

Manipulate self-organization: VSNs are often deployed as ad-hoc networks without any fixed infrastructure, requiring the sensor nodes to organize themselves into a cooperative network. Furthermore, they need to be able to pursue common goals with others and to adapt to the environment as it changes. Additionally, networks are often required to be self-healing such that they can recover from harm like failing nodes or malicious intruders. In general, the self-organization feature brings major challenges to the security of sensor networks, since nodes can typically solely rely on local sensor readings and on received information of others without having access to a global view of the network [59]. Attackers might aim at destroying the stability and proper functioning of a sensor network by manipulating its self-organization abilities. This is considered as tampering attack and would require an attacker to be an expert.

Fake/impersonate nodes (CWE-290, CWE-923, CWE-266, CWE-287): An attacker might want to steal other nodes’ identities or create fake nodes which do not exist in reality in order to fool benign nodes. This attack is considered as spoofing attack and requires an attacker to be skilled.

Selective transmission (CWE-417): An attacker might want to selectively communicate received data to other neighbors in order to disrupt the proper functioning of the application. A node that is the only connector of two network regions might want to not forward messages between the two regions to harm the application by splitting the network into two non-connected parts. This attack can harm the application and degrade services and is thus classified as tampering or Denial of Service attack. To launch a successful selective transmission attack, an attacker needs to be an expert.

Replication attacks (CWE-290, CWE-923, CWE-266, CWE-287): To perform node replication attacks, an attacker could copy the node ID of an existing node to add it to the sensor network. Using this, an attacker can participate in the application and perform a wider range of follow-up attacks including information eavesdropping or injection. Since the attack aims at impersonating nodes, it is considered as spoofing attack performed by an expert attacker with deep insight into the application.

AI bias attacks (CWE-840, CWE-20): As with the single-node center AI bias attacks, an attacker would want to influence AI algorithms to return faulty results. In this case, the expert attacker aims at manipulating the data flow between nodes which is used, e.g., for online learning.

### 4.4. Privacy Attacks (Cwe-359)

Attacks on privacy can happen on multiple layers. The ultimate goal of the attacker is to gain access to sensitive information on persons or objects observed by VSN nodes. Especially in highly sensitive areas like hospitals or private homes, this is an important category of attack. In general, attackers need to be at least skilled to perform a privacy-invasive attack.

On the physical layer: With the necessary technical skills and equipment, an attacker could physically compromise the sensor nodes and obtain the data and other key material. Additionally, an attacker can succeed in performing a side-channel attack to analyze the physical activities of the system to extract the cryptographic keys [21].

On the software layer: With access to the node’s local software, the attacker may extract privacy-violating information at the point of its processing.

On the network layer: Using a person-in-the-middle approach, an attacker could gain access to sensitive information.

On the middleware layer: A malicious node could subscribe to any topic in order to receive privacy relevant data such as images or processed results from all publishers in the network.

On the distributed application layer: Attackers could access the output of the distributed application including images and processing results such as interpreted scenes or behavior. Gathering processed information from VSNs is especially problematic in applications like industrial automation (manipulate the manufacturing process) or applications centered around humans (e.g., elderly care or assisted living) since they access privacy-relevant data.

### 4.5. On Repudiation Attacks

The proposed general threat model does not list any repudiation attacks. Those attacks are historically user-centered in the sense that a user of a system cannot deny the reception of a certain message. In a VSN, this problem can only arise in special contexts such as a network with user interaction or with high accountability requirements.

## 5. What Is Next? Process and Mitigation

To properly secure an IT-based system, security must be seen as a process that runs in parallel and is very interleaved with the development of the system. Threat modeling should be used as basic technique to continuously assess where a potential attacker could penetrate the system. As already mentioned above, this paper presents a generalized threat model for visual sensor networks. The threats described here are generalized in a sense that they will be present in most instances. While it is a good guideline, each specific sensor network instance should have a specific threat model that accounts for additional threats specific to this instance. STRIDE is a well-known technique which can be used for threat modeling, however, there are various different methods such as attack trees [71].

### 5.1. Scoring and Prioritizing

Identified threats and vulnerabilities should be ranked by their severity and addressed by their priority. For this purpose, there exist several scoring techniques which assess the risk that a specific threat poses to the system. One example is a technique called DREAD [72]. DREAD stands for Damage, Reproducibility, Exploitability, Affected users and Discoverability which are the categories used to assess the potential impact of a system flaw. For each category, subscores are assigned to the threats from which a total score is calculated. This score can then be used to prioritize amongst multiple threats in mitigation. Another widely-used scoring technique for vulnerabilities in a system is Common Vulnerability Scoring System (CVSS), which is presented in more detail in the following subsection.

#### Common Vulnerability Scoring System

The Common Vulnerability Scoring System (CVSS) is the most common scheme to quantify vulnerabilities in computing systems defined in an open standard (https://www.first.org/cvss/). It is a proven way of assigning scores to vulnerabilities by categorizing their preconditions and effects. It has been constantly updated over the past years and even specialized scoring systems for certain domains like robotics [73] have been derived from it. The latest version is CVSS v3.1. The CVSS base score takes two sets of metrics into account for each vulnerability: Exploitability and impact as detailed in Table 2. The value range for CVSS scores is between 0 and 10.

A CVSS score for a certain vulnerability can be encoded using a CVSS vector that names the value for each metric. As an example, the CVSS vector (AV:P/AC:L/PR:N/UI:N/S:U/C:N/I:N/A:H) receives an overall score of 4.6 out of 10. This vulnerability would require physical access (AV:P), has a low attack complexity (AC:L), needs no privileges to be exploited (PR:N), needs no interaction by the user (UI:N), does not go beyond the components security scope (S:U), has no impact on confidentiality (C:N) or integrity (I:N) but greatly impacts the availability of the component (A:H). This CVSS vector could describe the score for a vulnerability that equals to physical destruction of a component.

### 5.2. Mitigation Techniques

When vulnerabilities or potential weaknesses are noticed, a suitable mitigation must be found in order to proof the system against that. Cryptography is often the first mechanism which is considered to provide security. However, cryptography is typically only a part of the solution. A secure system also needs to consider the overall security architecture, operational processes and the application environment. In the end, a security system is only as strong as its weakest link. The best cryptographic infrastructure is of no benefit if the key material can easily be obtained by social engineering. In any case, a holistic view on the security of a system is recommended. Especially in a cyber–physical system like a visual sensor network, security considerations before deployment and a continuous monitoring are crucial.

Since solutions to certain vulnerabilities are always specific to the system, only general suggestions on the mitigation can be given in the context of this work. To find a proper mitigation for a certain threat, the weakness’ respective CWE number (which can be found in Section 4) is a good start. Table 3 summarizes the most important techniques that can be used to address STRIDE threats. It also gives some examples of their use in the sensor networks field. The digital mitigation techniques which can be used to secure VSNs against spoofing include authentication, data signing and data hashing. In many applications it is important to know by whom the data was produced, which can be achieved by authentication procedures exploiting, i.e., watermarking or digital signatures. Tampering in VSNs can be tackled through message authentication codes, tamper-resistant protocols and digital signatures. It is usually difficult to prevent tampering, however, it should at least be detected. One approach to detect the manipulation of images or videos is using checksums and digital signatures. Mitigation techniques against repudiation include blockchains, smart contracts, timestamping and encryption. One strategy is to store decentralized authentication and node trust information using a blockchain data structure. Information disclosure can be tackled via authorization and encryption procedures in order to ensure confidentiality of data. Denial of Service is in many cases difficult to prevent, especially when it comes to physical destruction of devices or jamming of the communication. However, systems should at least be able to detect DoS attacks, which can be done using Quality of Service, filtering and DoS-resistant protocols. QoS for example measures the transmission quality and service availability of a network. The standard strategy to tackle elevation of privileges attacks is to rely on the principle of least privilege. Doing so, a user or process are only given those privileges which are essential to perform its intended function. More in-depth descriptions and overviews can be found in security-surveys like [10,20,32,74].

## 6. Conclusions

In this paper, we have collected the most common threats, in form of a strategical threat model for visual sensor networks. We have presented a structured analysis of potential threats to each layer of such networks from physical via the node-level up to the network, middleware and distributed application layer.

This generalized threat model is meant to be a guide in performing threat modeling for a specific application. In applications, security engineering should be a process where threat modeling, penetration testing and mitigation are done continuously. We have described how to deduct threats in a structured security process and how to proceed once threats or vulnerabilities have been identified.

Future work could include an extension of the threat model to also consider heterogeneous network setups and next generation technologies such as mirco- and nanoscale camera devices for in-body applications. Furthermore, future work should also be directed towards the establishment of a workflow which supports engineers in considering and maintaining security aspects from the very beginning of the application development. One example for such a workflow is DevSecOps (https://devops.com/) (Development, Security, Operations), which is a practice to better align security, engineering and operations by infusing security throughout the development and operations lifecycle of applications. Thus, DevSecOps aims at embedding security in every part of the development process. Such a model can be used as a basis to establish such a workflow for VSN applications. Finally, an important future work task would be to extend existing taxonomies like CWE to cyber–physical systems, since up until now they do not typically consider physical attacks.

## Figures and Tables

**Figure 1 sensors-20-03629-f001:**
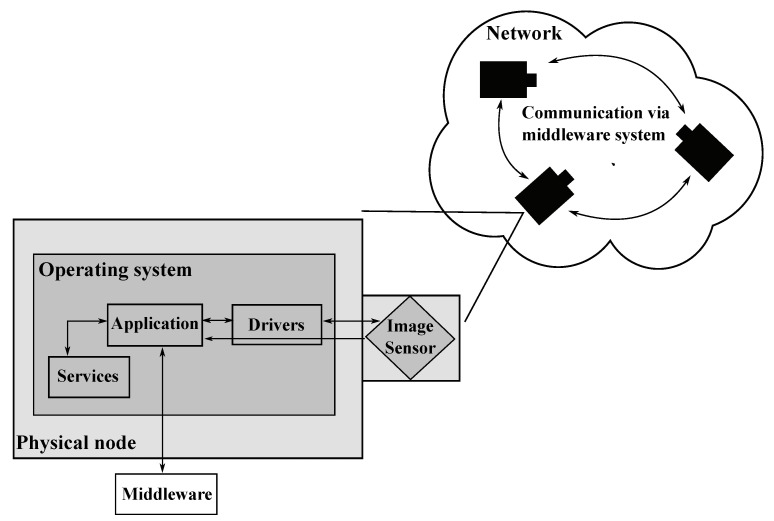
Layers of a networked visual sensor node which are prone to different types of attack.

**Figure 2 sensors-20-03629-f002:**
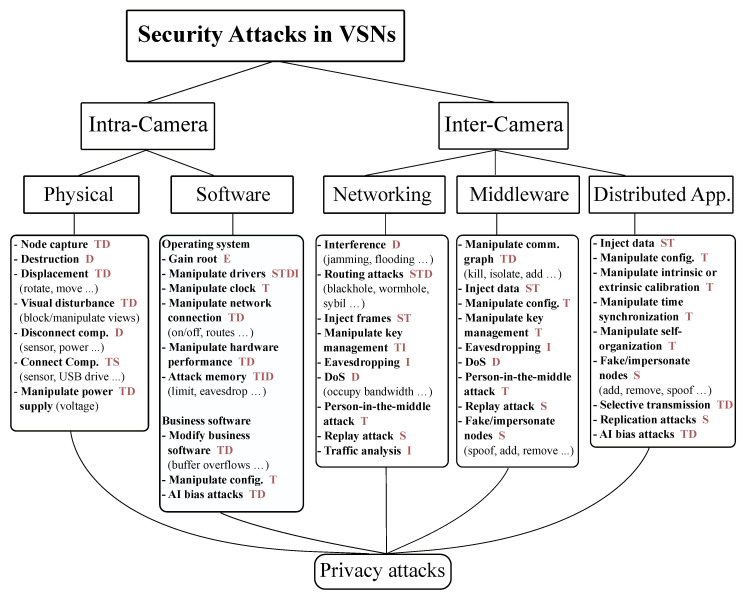
Threat model for visual sensor networks (VSNs), which categorizes the attacks into two main categories, namely intra-camera and inter-camera. Besides each attack, the corresponding STRIDE categorization is shown.

**Figure 3 sensors-20-03629-f003:**
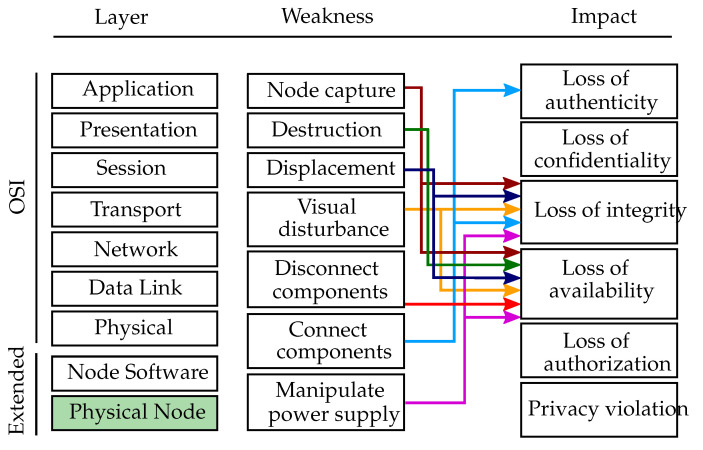
Physical layer weaknesses. Overview of the physical layer weaknesses mapped to the affected extended-OSI layer (green colored box) and the respective threat impacts.

**Figure 4 sensors-20-03629-f004:**
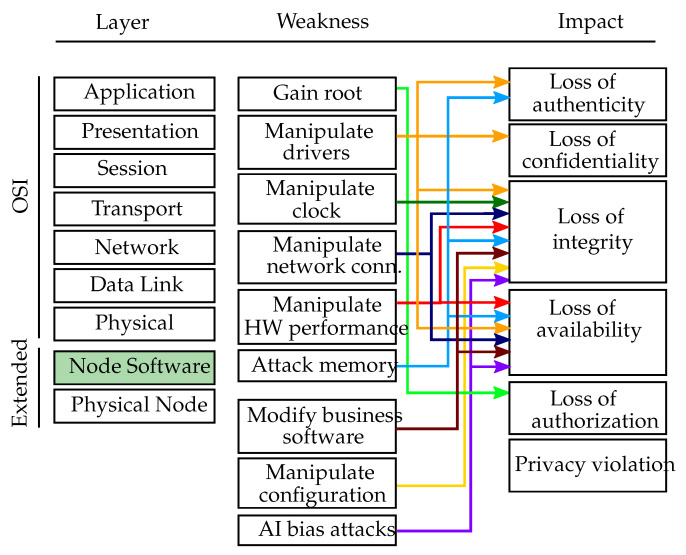
Node software layer weaknesses. Overview of the software layer weaknesses, including the operating system and the business software, mapped to the affected extended-OSI layer (green colored box) and the respective threat impacts.

**Figure 5 sensors-20-03629-f005:**
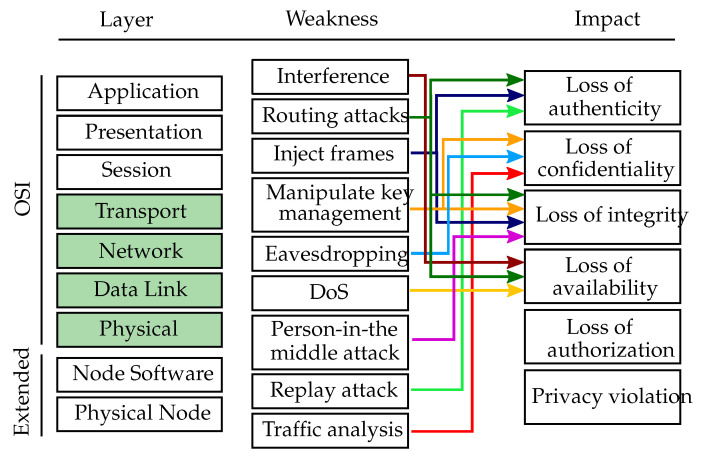
Network layer weaknesses. Overview of the networking layer weaknesses mapped to the affected OSI layers (green colored boxes) and the respective threat impacts.

**Figure 6 sensors-20-03629-f006:**
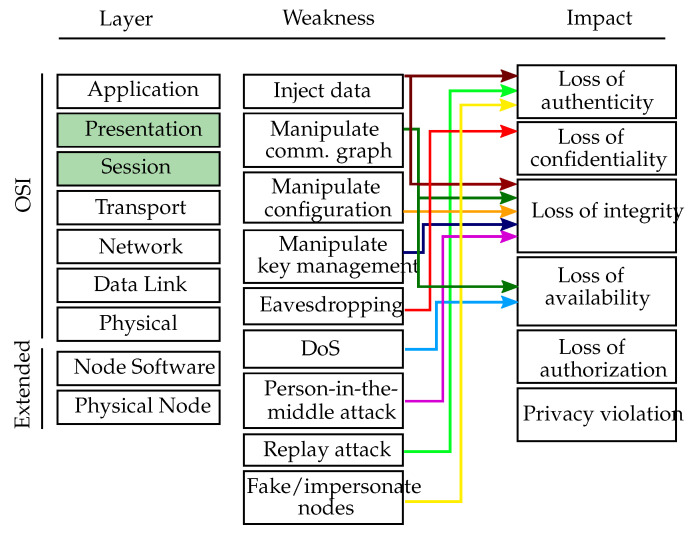
Middleware layer weaknesses. Overview of the middleware layer weaknesses mapped to the affected OSI layer (green colored boxes) and the respective threat impacts.

**Figure 7 sensors-20-03629-f007:**
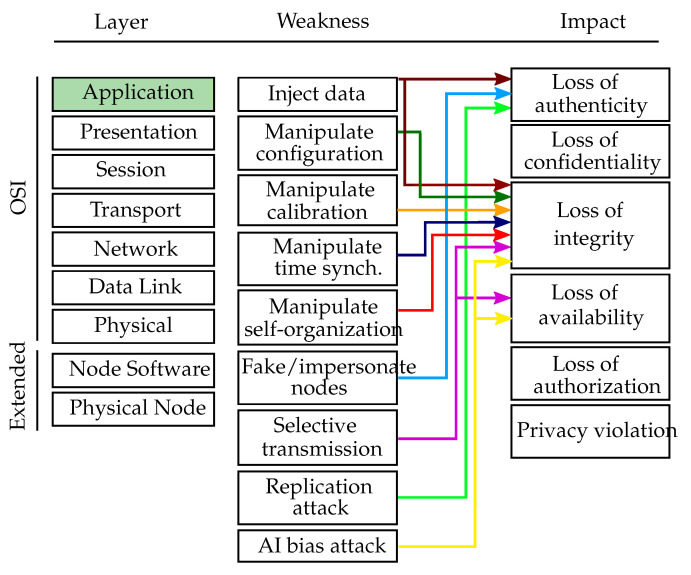
Distributed application layer weaknesses. Overview of the distributed application layer weaknesses mapped to the affected OSI layer (green colored box) and the respective threat impacts.

**Table 1 sensors-20-03629-t001:** The STRIDE threats.

Threat	Definition	Violated Property
Spoofing	Impersonating something or someone else	Authenticity
Tampering	Modifying data or code	Integrity
Repudiation	Claiming to have not performed an action	Non-repudiation
Information Disclosure	Exposing information to someone who is not authorized to see it	Confidentiality
Denial of Service	Deny, degrade or interrupt any services	Availability
Elevation of Privilege	Gain capabilities without authorization	Authorization

**Table 2 sensors-20-03629-t002:** The basic Common Vulnerability Scoring System (CVSS) metrics. Each qualitative factor for a metric (e.g., low) has a dedicated multiplier defined in CVSS that will then be used to calculate the CVSS score.

Category	Metric	Potential Values	Comment
Exploitability	Attack vector (AV)	physical (P), local (L), adjacent network (A), network (N)	Names the attack vector for the vulnerability, i.e., if an attacker needs to be physically at the target system, needs access to the local system, needs to be in an adjacent network (e.g., Wifi) or if the vulnerability is remotely exploitable
Exploitability	Attack Complexity (AC)	low (L), high (H)	Details how complex the vulnerability would be to exploit
Exploitability	Privileges Required (PR)	none (N), low (L), high (H)	Details if the attacker requires special privileges to exploit a vulnerability
Exploitability	User Interaction (UI)	none (N), required (R)	Describes if the legitimate user of the system needs to take an action for an attack to be successful
Exploitability	Scope (S)	unchanged (U), changed (C)	Describes if an exploit of the vulnerability allows to affect components beyond the scope of the vulnerable component
Impact	Confidentiality Impact (C)	none (N), low (L), high (H)	Describes the potential loss of confidentiality in the vulnerable component (i.e., if the attacker can access data)
Impact	Integrity Impact (I)	none (N), low (L), high (H)	Describes the potential loss of integrity of the vulnerable component (e.g., if the component remains trustworthy despite an attack)
Impact	Availability Impact (A)	none (N), low (L), high (H)	Describes the impact on the vulnerable component’s availability

**Table 3 sensors-20-03629-t003:** An overview of mitigation techniques.

STRIDE	Digital Mitigation Techniques Usable in VSNs	Examples
Spoofing	Authentication	[29,75]
	Data signing	
	Data hashing	
Tampering	Message authentication codes	[29,57,76]
	Tamper-resistant protocols	
	Digital signatures	
Repudiation	Blockchains	[77,78,79]
	Smart contracts	
	Timestamping	
Information disclosure	Authorization	[29,64]
	Encryption	
Denial of Service	QoS	[54,56,61,62,80,81]
	Filtering	
	DoS-resistant protocols	
Elevation of privileges	Principle of least privilege	[29]

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
