# Peer review of "A Generalized Threat Model for Visual Sensor Networks"

_sensors, 2020, doi:10.3390/s20133629_

Round 1
Reviewer 1 Report
The paper is in general well-written and presents a comprehensive study on vulnerabilities in VSNs.
I believe the threat model presented by the authors is sound.
I consider the paper is almost ready for publication, besides some minor issues that can be easily fixed:
1) Please, avoid using sentences in the first person (we, our, us), prefer using the third person in scientific text.
2) Subsections only make sense to use if you want to divide contents if you have just one subsection, it means that you don’t need it. So, fix this problem in the introduction, you don’t the division of that subsection 1.1. The same observation is valid for Section 2.
3) The literature review in Section two is fine. However, I missed a discussion of heterogeneous networks, such as in: "Improving network resources allocation in smart cities video surveillance", "Computer Networks", 2018. doi = "https://doi.org/10.1016/j.comnet.2018.01.042",
Please, include this reference and provide a small discussion of the case in which such a heterogeneous network setup is present. Maybe you even enhance your threat model considering this possibility, but this is not mandatory, as you can try to fit it in your current model.
4) Elaborate more in the text describing the attacker model in Section 3.3.
5) I think Section 4 could be more pedagogical if the authors used figures to illustrate the text. Please, check the Open Web Application Security Project (OWASP) Top-10 Security Risks for examples of how you could build illustrative figures that would enhance much your paper!
6) Section 5.2 is also too summarized. Please, elaborate more on the possible Mitigation Techniques.
7) Include a final paragraph in the conclusion section to briefly discuss future works.
Reviewer 2 Report
Authors have proposed a threat model for visual sensor networks. The topic is interesting and worth investigation. However there are some issues that need to be addressed before getting published.
- There are some typos and grammatical issues in the manuscript.
- Organisation of the article need some modifications, i.e., Motivation should be a separate section.
- Overview of all the sections are missing that should be added at the end of introduction.
- It is always recommended to use full name instead of acronym as a title or subtitle like IoT, SNs.
- Contributions should be clearly mentioned.
- I would also strongly recommend adding a comparison of existing models and evaluation of the proposed model to understand the strength and weakness.
- Authors are also asked to consider OWASP and/or MITRA/NIST vulnerability list in their model and mention that they have followed their guideline.
- A clear figure is required to show the matching of each OSI layer with structured analysis of potential threats.
- It is also not clear how they are using cryptography in their model and what may be the specific purpose. They have mentioned only Repudiation.
